# Association of Excessive Daytime Sleepiness with the Zung Self-Rated Depression Subscales in Adults with Coronary Artery Disease and Obstructive Sleep Apnea

**DOI:** 10.3390/diagnostics11071176

**Published:** 2021-06-28

**Authors:** Yeliz Celik, Hale Yapici-Eser, Baran Balcan, Yüksel Peker

**Affiliations:** 1School of Medicine, Koç University Research Center for Translational Medicine (KUTTAM), Istanbul 34450, Turkey; hyapici@ku.edu.tr (H.Y.-E.); yuksel.peker@lungall.gu.se (Y.P.); 2Graduate School of Health Sciences, Koç University, Istanbul 34450, Turkey; 3Department of Psychiatry, School of Medicine, Koç University, Istanbul 34450, Turkey; 4Department of Pulmonary Medicine, School of Medicine, Marmara University, Istanbul 34722, Turkey; drbaranbalcan@gmail.com; 5Department of Clinical Sciences, Respiratory Medicine and Allergology, School of Medicine, Lund University, 22185 Lund, Sweden; 6Department of Molecular and Clinical Medicine/Cardiology, Institute of Medicine, Sahlgrenska Academy, University of Gothenburg, 40530 Gothenburg, Sweden; 7Division of Pulmonary, Allergy, and Critical Care Medicine, University of Pittsburgh School of Medicine, Pittsburgh, PA 15213, USA

**Keywords:** obstructive sleep apnea, excessive daytime sleepiness, depression, Zung SDS, coronary artery disease

## Abstract

Excessive daytime sleepiness (EDS) is a factor associated with both obstructive sleep apnea (OSA) and depressive symptoms. Continuous positive airway pressure (CPAP) treatment may decrease EDS in adults with OSA; however, the modulatory role of depressive symptoms on the improvement of EDS is not known. We aimed to explore the association between subscales of the Zung Self-rated Depression Scale (SDS) and Epworth Sleepiness Scale (ESS) over a 2-year period in coronary artery disease (CAD) patients with OSA. This was a post-hoc analysis of the RICCADSA cohort, in which 399 adults with CAD (155 sleepy OSA [apnea–hypopnea index ≥ 15/h] and ESS score ≥ 10, who were offered CPAP; and 244 nonsleepy OSA [ESS < 10]), randomized to CPAP [*n* = 122] or no-CPAP [*n* = 122]) were included. Three factors were extracted from the Zung SDS, based on the principal component analysis: F1, cognitive symptoms and anhedonia; F2, negative mood; and F3, appetite. In a mixed model, the ESS score decreased by 3.4 points (*p* < 0.001) among the sleepy OSA phenotype, which was predicted by the decline in the F2, but not in the F1 and F3 scores. The fixed effects of time were not significant in the nonsleepy OSA groups, and thus, further analyses were not applicable. Additional within-group analyses showed a significant decrease in all subscales over time both in the sleepy and nonsleepy OSA patients on CPAP whereas there was a significant increase in the nonsleepy OSA group randomized to no-CPAP. We conclude that the improvement in negative mood symptoms of depression, but not changes in cognitive symptoms and anhedonia as well as appetite, was a significant predictor of decline in the ESS scores over a 2-year period in this CAD cohort with sleepy OSA on CPAP treatment.

## 1. Introduction

Obstructive sleep apnea (OSA) is a common sleep-related breathing disorder among adults with coronary artery disease (CAD) [1]. Observational studies have suggested that OSA may increase the risk of incident CAD and mortality [1]. OSA may induce excessive daytime sleepiness (EDS), which is the most common treatment indication for continuous positive airway pressure (CPAP). In OSA patients, EDS is shown to be associated with higher health care use for hypertension, diabetes and depression [2]. Moreover, a significant number of patients with OSA experience residual EDS despite an adherent use of CPAP, and the reasons for the residual EDS are not well defined [3,4].

OSA may also cause depression by disturbing sleep patterns and through its cardiovascular effects [5]. In this context, the evaluation of depression in cardiac patients with concomitant OSA is challenging due to the difficulty of distinguishing depressive symptoms from EDS and the multidirectional relationship of EDS, OSA, CAD and depression.

We have recently demonstrated significant reductions in Zung Self-rating Depression Scale [SDS] scores in nonsleepy OSA patients randomized to CPAP compared with those in the no-CPAP group in the RICCADSA trial [6]. Moreover, there was an even higher benefit in the sleepy OSA phenotype following one year of CPAP treatment [7], suggesting that improvement in depression is strongly dependent on the improvement in EDS. Furthermore, EDS scores are also significantly correlated with depression scores [5,6], and improvement of EDS scores is shown to be modulated by the improvement of depressive symptoms [5,6,7,8,9,10]. A better understanding of this bidirectional relationship is therefore needed.

The relationship between EDS and depression may depend mostly on symptoms such as fatigue and changes in psychomotor activity as well as sleep patterns. Depression scales include heterogeneous symptom profiles of somatic, cognitive and affective categories.

In the current study, we aimed to explore the association between different symptom groups of depression and EDS over a 2-year period, which may have prognostic and therapeutic implications in this context.

## 2. Materials and Methods

### 2.1. Study Population

This was a secondary analysis of the RICCADSA trial. The RICCADSA study population has been previously described elsewhere [11]. In brief, adults with a history of percutaneous coronary intervention (PCI) or coronary artery by-pass grafting (CABG) within 6 months prior to study start were invited to participate in the *R*andomized *I*ntervention with *C*PAP in *C*oronary *A*rtery *D*isease and Obstructive *S*leep *A*pnea (RICCADSA) trial. The patients were enrolled into the study between 2005 and 2010, and the follow-up for the primary outcomes was ended in May 2013 [12].

As illustrated in Figure 1, 399 patients met the inclusion criteria and were included in the current subanalysis. The participants were classified as OSA (apnea-hypopnea index [AHI] ≥ 15/h) and no-OSA (apnea-hypopnea index [AHI] < 5/h) based on the baseline sleep recordings with cardiorespiratory polygraphy. The cut-off for AHI (15/h) was chosen for the OSA diagnosis, given that the diagnostic procedure at baseline was based on cardiorespiratory sleep studies, not on gold standard polysomnography, and thus, the total sleep time could not be given exactly. This cut-off value was previously shown to be reliable for the OSA diagnosis whereas the cut-off AHI 5/h was reliable for excluding OSA [13]. Patients with AHI 5.0–14.9/h were considered as a borderline group, and were not included in the main trial (Figure 1). The individuals with nonsleepy OSA phenotype (Epworth Sleepiness Scale [ESS] < 10) were assigned to CPAP or no-CPAP, and the sleepy OSA phenotype (ESS ≥ 10) received CPAP.

The RICCADSA trial protocol was approved by the Regional Ethical Review Board in Gothenburg (approval nr 207-05; 2005 September 13; amendment T744-10; 2010 November 26; amendment T512-11; 2011 June 16), and all patients provided written informed consent [12]. The trial was registered with ClinicalTrials.gov (NCT 00519597).

### 2.2. Cardiorespiratory Polygraphy

Baseline sleep studies were conducted by using the Embletta^®^ Portable Digital System device (Embla, Broomfield, CO, USA). As previously described in detail [11], the Chicago criteria were used for apnea and hypopnea definitions [14].

### 2.3. Epworth Sleepiness Scale

The ESS questionnaire [15] contains eight questions regarding the probability of dozing off under eight different situations. The maximum score is 24, and a cut-off value of 10 was used to define the EDS in the RICCADSA trial.

### 2.4. Zung Self-Rating Depression Scale

The Zung SDS is a well-recognized questionnaire, that gives both a total score, and a categorical rate of depressive mood [16]. In brief, the scale contains 20 questions (Table 1) with the self-rating scale ranging from 1 to 4 points (1 = a little of the time, 2 = some of the time, 3 = good part of the time, 4 = most of the time). All participants were requested to complete the Zung SDS questionnaire at baseline and at 1 and 2-year follow-ups. As the item 4 (“I have trouble sleeping at night”) is a sleep-related question, it was excluded from the factor analysis. Three factors are extracted based on the principal component analysis; F1, cognitive symptoms and anhedonia; F2, negative mood; and F3, appetite (Table 1).

### 2.5. Group Assignment, Randomization, Interventions and Follow-Up

As illustrated in Figure 1, all participants with sleepy OSA phenotype as well as the nonsleepy OSA phenotype, who were randomized to CPAP, were provided with a self-titrating CPAP device (S8^®^, or S9^®^; San Diego, CA, USA) with a nasal or full-face mask and humidifier. All participants receiving CPAP were contacted by telephone after one week, and all patients were followed at 1 year and 2 years [11].

### 2.6. Adherence to CPAP

All patients receiving CPAP brought their devices to the clinic at each follow-up visit, CPAP hours/night and CPAP days/period were downloaded from the device, and accumulated CPAP hours/night/all nights were calculated. All technical adjustments were done according to the clinical routines by the sleep medicine unit staff.

### 2.7. Statistical Analysis

The study sample distribution of demographics and clinical characteristics at baseline was examined using the descriptive statistics. Continuous variables were reported as mean ± standard deviation, or median values with boundaries of the interquartile ranges (IQR), and the categorical variables as numbers and percentages. The chi-square test was used to compare the subgroups on the categorical variables, and for the within-group changes over time, nonparametric tests were used for the continuous variables (median with IQRs).

A principal component analysis (PCA) was conducted on the Zung SDS with orthogonal rotation (VARIMAX). The Kaiser-Meyer-Olkin Measure was used to test the sample adequacy. Bartlett’s test was applied to check for the sphericity assumption. The primary item inclusion was set at 0.40 (absolute value). The number of factors were decided based on the Kaiser’s Criterion (eigen values >1.0) and the interpretability of resulting factor structures.

A linear mixed model was applied to examine individual changes on the estimated mean ESS scores in response to CPAP treatment among the CAD patients with OSA based on the procedure described by Heck et al. [17]. As illustrated in Figure 2, a two-level growth modeling approach was used to test the effect of “time” as the effects of CPAP treatment on the ESS. ESS scores, as the outcome, were measured three times for each patient to examine the pattern of change over the temporal period of study. The individual change was considered as the expected within-subjects effect for (level 1) whereas the shape of growth trajectory (time-related slope) was considered as the expected between-subjects effect (level 2). The “time” as a linear trend, coded as 0, 1 and 2, was entered into the model as a fixed effect as well as random effects for the intercept and slope. The covariance matrix of the random effects was assumed to be unstructured. The “time” as a quadratic trend, coded as 0, 1 and 4, was also added to the model as the fixed effect. The scores of F1, F2 and F3 were considered between-subject predictors. They were included to the mixed model as the fixed effects to explain possible differences in their individual intercepts and time-related slopes. Moreover, age, body-mass-index (BMI), AHI, CPAP use (hr/night) and comorbidities were tested for possible main effects and interactions. The statistical procedure was performed in the sleepy OSA group on CPAP treatment, and nonsleepy groups randomized to CPAP v.s. no-CPAP, respectively. All statistical tests were two-sided, and *p* < 0.05 was considered statistically significant. Statistical analysis was performed using the Statistical Package for Social Sciences, version 26.0 for Windows^®^ system (SPSS^®^ Inc., Chicago, IL, USA).

## 3. Results

### 3.1. Principle Component Analysis

A principal component analysis was conducted on the 19 items with orthogonal rotation (VARIMAX). The Kaiser-Meyer-Olkin measure verified the sampling adequacy for the analysis, KMO = 0.884. Bartlett’s test of sphericity χ^2^ (190) = 2409.6, *p* < 0.001, indicated that correlations between items were sufficiently large for the PCA. Initially, an analysis was run to obtain eigenvalues for each factor in the data. Five factors had eigenvalues over 1 (Kaiser’s criterion) and they explained 52.7% of the variance in the data. The scree plot was slightly ambiguous and showed inflexions. Therefore, another PCA was conducted by setting the number of factor extractions by 3. Table 1 shows the factor loading after rotation. Items 2 and 8 were excluded due to item inclusion criteria (> 40). The items that cluster on the same factors suggest that factor 1 represents cognitive symptoms and anhedonia, factor 2 negative affective symptoms and factor 3 appetite.

### 3.2. Baseline Characteristics of the Study Population

As shown in Table 2, the sleepy OSA patients were slightly younger and more obese and had higher AHI and ODI than the patients with non-sleepy OSA. Baseline comorbidities were similar.

### 3.3. Within-Group Differences Over Time

As expected, there was a significant decrease in ESS scores over time among the sleepy OSA phenotype on CPAP (Figure 3). Interestingly, the decline in ESS scores was observed even in the nonsleepy phenotype on CPAP. ESS scores did not change significantly in the nonsleepy no-CPAP group. Average CPAP usage was 2.8 ± 2.9 h/night at 1 year, and 2.6 ± 2.9 h/night at 2 year follow-up in the nonsleepy group, and the corresponding values were 3.4 ± 2.6 h/night v.s. 3.3 ± 2.7 h/night in OSA patients with the sleepy phenotype.

To examine the linear trends on the Zung SDS subscales over time, the mean scores were computed among each group at baseline and follow-ups, respectively. As illustrated in Figure 4, there was a decreasing trend over time on the mean scores of F1 and F2 among both the sleepy and nonsleepy OSA patients on CPAP treatment (F1 14.2, 13.2, 13.4; F2 12.7, 11.2, 10.9; F3 3.5, 3.2, 2.9 in sleepy OSA (blue lines in a, b and c); and F1 13.9, 13.1, 13.3; F2 11.6, 10.3, 10.3; F3 3.8, 3.1, 3.2 in nonsleepy OSA (green lines in a, b and c), respectively) whereas an increasing trend was observed among the nonsleepy OSA patients randomized to no CPAP (F1 12.8, 14.0, 13.7; F2 10.7, 11.2, 10.6; F3 3.5, 3.3, 3.1; red lines in a, b and c). The mean scores were not compared since the normality assumption did not hold. For the within-group comparisons, the median values of the cognitive symptoms and anhedonia decreased from 14.0 (11.0–17.0) to 13.0 (10.0–16.0) after 1 year (*p* = 0.03), and to 13.0 (10.0–16.00) after 2 years (*p* = 0.07) among the sleepy patients on CPAP, and from 14.0 (10.0–16.5) to 13.0 (10.0–16.0) after 1 year (*p* = 0.04), and to 13.0 (9.0–16.0) after 2 years (*p* = 0.23) among the nonsleepy OSA patients on CPAP. On the contrary, the median scores increased from 11.5 (10.0–16.0) to 13.0 (10.0–17.0) after 1 year (*p* = 0.001), and to 13.0 (10.0–16.0) after 2 years (*p* = 0.03) in the no-CPAP group.

Corresponding median values for the negative mood were 12.0 (10.0–15.0) at baseline, 11.0 (9.0–13.0) at 1-year follow up, and 10.0 (9.0–12.0) at 2-year follow-up for the sleepy OSA phenotype on CPAP (*p* < 0.001 for both changes). The nonsleepy OSA phenotype on CPAP treatment demonstrated similar changes on the median scores, from 11.0 (9.0–13.0) at baseline to 10.0 (8.0–12.0) at both follow-ups (*p* < 0.001 for both). The median changes were not significant among the no-CPAP group.

Regarding the appetite subscale scores, there were significant within-group changes at 1-year and 2-year follow-ups compared to baseline in all subgroups (*p* = 0.04 vs. *p* = 0.001 among the sleepy OSA patients on CPAP, *p* < 0.001 v.s. *p* = 0.002 among the nonsleepy OSA patients randomized to CPAP, and *p* = 0.03 vs. *p* = 0.01 among the patients randomized to no-CPAP).

### 3.4. The Linear Mix-Model

#### 3.4.1. Null Model

In the null models, the mean for the ESS scores, including baseline, after 1 year and 2 years, were 10.2 in the sleepy OSA group who received CPAP, and 5.5 in the non-sleepy OSA patients who were allocated to CPAP, and 5.2 in the non-sleepy OSA patients who were randomized to no-CPAP (Table 3).

Within-individual variances (Level 1), which summarizes the population variability in average ESS scores around personal growth trajectory, was estimated around 9.9 for the sleepy OSA patients, 3.3 for the non-sleepy patients randomized to CPAP, and 3.3 for the non-sleepy patients with no CPAP, respectively. Between-individual variance (Level 2) was estimated 2.0 in the sleepy group on CPAP, 3.8 in the non-sleepy group on CPAP, and 4.1 in the non-sleepy group who were randomized to no CPAP (Table 3).

#### 3.4.2. The Shape of Growth Trajectory

The fixed effects are presented in Table 4. The estimate was 12.2 points for the intercept (*p* < 0.001), −4.00 points for the linear growth rate (*p* < 0.001) and 1.1 points for the quadratic growth rate (*p* < 0.001). These results indicate that the time-dependent polynomials were significant predictors for the change in ESS, and thus, both the linear and quadratic polynomials should be retained in subsequent analyses in sleepy OSA patients on CPAP. Therefore, it was decided that the time-related slopes were included to the next mixed model.

As presented in Table 4, the estimate of the intercept for ESS at baseline was 5.5 in the non-sleepy OSA group randomized to CPAP as well as in the no-CPAP group (*p* < 0.001 for both). However, the linear and quadratic polynomials were not significant predictors of mean ESS scores (Table 4), and convergence has not been achieved. Thus, there was no need to run further analyses in the nonsleepy OSA patients.

#### 3.4.3. Varying Time-Related Slope in the Sleepy OSA Patients on CPAP

As demonstrated in Table 5, the estimate for Level 1 was 4.9 (*p* < 0.001). At Level 2, random intercept for the between-individual variance was 2.1 (*p* = 0.02). Furthermore, the estimate for the random slope (linear polynomials) was 1.2, which also significantly varied between individuals *(p* = 0.04). However, the estimate of the covariance between the random slope and intercept was not statistically significant.

#### 3.4.4. Adding the Between-Subject Predictors

In order to explain variability between individuals, F1, F2 and F3 were included to the model, respectively, followed by the predictors age, gender, BMI, CPAP hours, AHI, ODI and comorbidities.

The fixed effects are presented in Table 6. The estimate of the intercept for ESS was 7.1 (*p* = 0.006) in the sleepy OSA group. The estimate for the fixed effect was 0.28 points for the F2 (*p* < 0.001), −2.88 points for the linear growth rate (*p* < 0.001), and 0.6 points for the quadratic growth rate (*p* = 0.03). For the rest of the predictors, the mixed model showed no significant associations, and these predictors were therefore removed from further analyses.

#### 3.4.5. Final Mixed Models in Sleepy OSA Patients on CPAP

The estimate of the intercept for ESS was 8.9 points, 0.26 points for the F2 (*p* = 0.001). The time related polynomials were −3.4 and 0.87 points, respectively (*p* < 0.001 for both). The within-individual variance was 4.67 points (*p* < 0.001) while the random intercept for the between-individual variance was 2.16 points (*p* = 0.01) (Table 7).

## 4. Discussion

The main finding of the current study was that improvement in negative mood was significantly associated with improvement in EDS over a 2-year period on CPAP among the CAD patients with sleepy OSA at baseline. The changes in cognitive distortions and anhedonia as well as in appetite were not related. Previous reports were focused on the association between total depression scores and EDS [18]. More recently, we have demonstrated the beneficial effects of CPAP on depression in the RICCADSA cohort, especially among the sleepy OSA phenotype [7]. The current paper revealed that the improvement in EDS was mainly based on the improvement in negative mood independent of age, BMI, AHI at baseline and CPAP use (h/night).

This finding is unique as it shows that the overlap of depression and EDS is mainly through negative mood, instead of anhedonia as the loss of ability to get joy from activities, cognitive distortions and thought content as feeling useless. It is reasonable that a patient with negative mood may be less likely to be adherent to CPAP treatment and get less improvement in EDS as opposed to a non-depressed patient. It is also plausible that adding an intervention (i.e., CPAP in this case) may improve negative mood via a placebo effect. Notwithstanding, adding interventions to improve mood in CAD patients with sleepy OSA may promote additional benefits in improvement in EDS, probably by concomitant improvement in adherence to CPAP treatment.

As extensively discussed in our previous publications, EDS is one of the most important symptoms of OSA, and there seems to exist an interaction between EDS and depression in OSA patients (6,7). The beneficial effects of CPAP treatment for EDS are already well-known [19]. We demonstrated improvements in both SDS and ESS scores following 1 year of CPAP treatment in both phenotypes [7]. Moreover, there was a significant relationship between CPAP use in h/night and improvement in mood in multivariate analysis adjusted for confounding factors, and the cut-off value of CPAP usage for improvement in mood was 4 h/night when the ESS score change from baseline was entered into the model [7].

In the current study, we did not perform subanalyses within the CPAP usage categories since our aim was to explore the association between the SDS subscales and EDS, in which CPAP usage was included as one of the predictors in the final model. However, it was not a significant predictor in this subgroup of patients with the sleepy OSA phenotype. Though not related to the improvement in EDS, there was a significant decrease in cognitive symptoms and anhedonia scores as well as appetite scores among OSA patients who were treated with CPAP. These results suggest that our previous reports, showing a significant decrease in the total Zung SDS scores (6,7) were covering all subscales, though only the changes in negative mood were significantly associated with the changes in EDS among the sleepy OSA phenotype.

The reduction in appetite scores (increased appetite and weight gain) in the whole cohort was indeed interesting. This might explain a natural course of reduced depressive mood during the recovery period following a revascularization. This may also partly explain weight gain observed among patients with OSA following CPAP treatment [20].

To the best of our knowledge, association of sub-dimensions of depression with OSA as well as with EDS has yet not been studied in neither general population nor in sleep clinic cohorts, and nor in cardiac populations. The extent of the decline in negative mood and cognitive symptoms scores was significantly greater among both sleepy and nonsleepy OSA patients who were on CPAP treatment compared to CAD patients with untreated OSA. Taken together, improving a patient’s EDS appears to be crucial for improvement in negative mood in CAD patients with OSA. Similarly, adding interventions to improve mood may improve EDS in CAD patients with OSA.

The current study has several strengths and some limitations of note. To the best of our knowledge, this is one of the largest existing studies in CAD patients with OSA on CPAP treatment addressing the association between sub-dimensions of depression and EDS changes over a 2-year period. As previously shown in a meta-analysis [21], the occurrence rates of depression range from 5% to 47% in cardiac patients. It has been suggested that these differences may reflect the various methodological instruments used in the evaluation of depression [22]. The National Heart, Lung, and Blood Institute (USA) had therefore recommended an investigation to address whether or not depression questionnaires used in the general population were applicable to cardiac patients [23]. In this context, a study by Perez Adel et al. [24] supported the use of the Zung SDS scores with justifications such as (*A*) it was a widely used self-questionnaire [25]; (*B*) it was easy to understand [26]; and (*C*) it predicted mortality and new hospitalizations in cardiac failure patients [27] as well as the risk for cardiovascular death in adults free of CAD at baseline [28]. It has also been argued that the use of total scores of a questionnaire may be problematic since different clusters of symptoms might be underestimated [23], and it might be difficult to diagnose and treat various types of depression. Here, by conducting a two-level linear mixed model, we could explore the association between the subscales of Zung SDS and EDS changes over a 2-year period. The long-term follow-up is an additional strength of the study. We should also acknowledge our study limitations. Firstly, the depression diagnosis was not validated with a structured clinical interview. Since the analyses are mostly related to self-reported questionnaires, we cannot exclude inherent biases. Secondly, sleepy and nonsleepy OSA patients were categorized based on the ESS threshold, which may not be precise. However, this questionnaire is a largely accepted tool used in clinical cohorts. Finally, the patients with sleepy OSA phenotype were more obese and had more hypoxemia than the patients with nonsleepy OSA at baseline. However, these variables were included in the multivariate analysis as covariates but they were not significant predictors of EDS in this cohort.

## 5. Conclusions

We conclude that improvement in negative mood symptoms of depression, but not changes in cognitive symptoms and anhedonia as well as appetite, was associated with the improvement in EDS in the sleepy RICCADSA cohort, who were treated with CPAP. Moreover, the magnitude of decline in negative mood and cognitive symptoms scores was significantly greater among both sleepy and nonsleepy OSA patients who were on CPAP treatment. Thus, improving a patient’s EDS seems to be important to achieve improvement in negative mood in CAD patients with sleepy OSA. Likewise, adding interventions to CPAP treatment of OSA to improve mood may promote CPAP adherence, and, similarly, better adherence to CPAP treatment and alleviation of the underlying OSA may improve mood in patients with CAD.

## Figures and Tables

**Figure 1 diagnostics-11-01176-f001:**
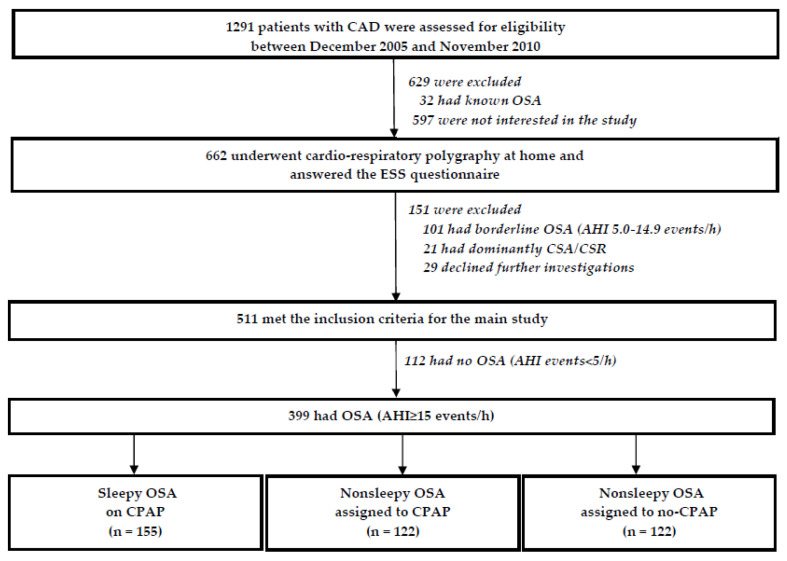
Flow-chart of the study population. AHI, apnea–hypopnea index; CAD; coronary artery disease; CPAP, continuous positive airway pressure; CSA-CSR, central sleep apnea–Cheyne Stokes respiration; ESS, Epworth Sleepiness Scale; OSA, obstructive sleep apnea.

**Figure 2 diagnostics-11-01176-f002:**
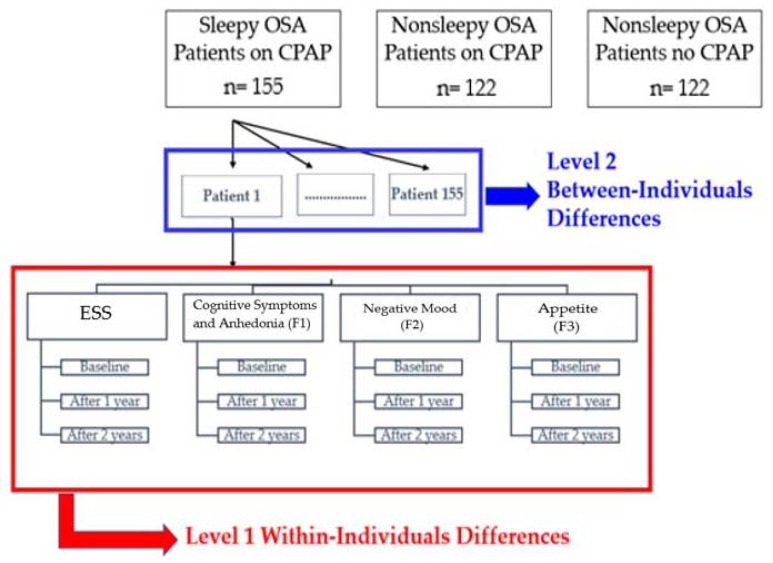
A two-level linear mixed model of the study design. CPAP, continuous positive airway pressure; ESS, Epworth Sleepiness Scale; OSA, obstructive sleep apnea.

**Figure 3 diagnostics-11-01176-f003:**
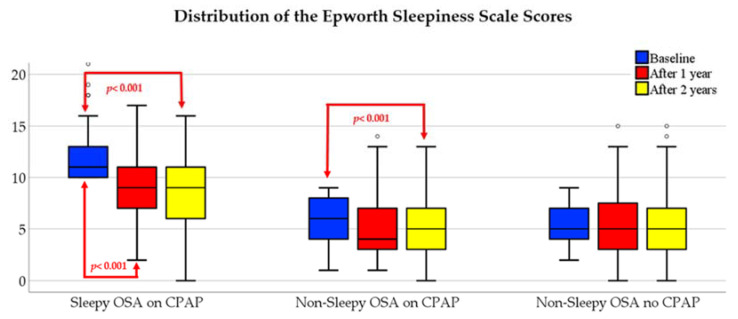
Changes in the Epworth Sleepiness Scale scores over time in the study population. CPAP, continuous positive airway pressure; OSA, obstructive sleep apnea.

**Figure 4 diagnostics-11-01176-f004:**
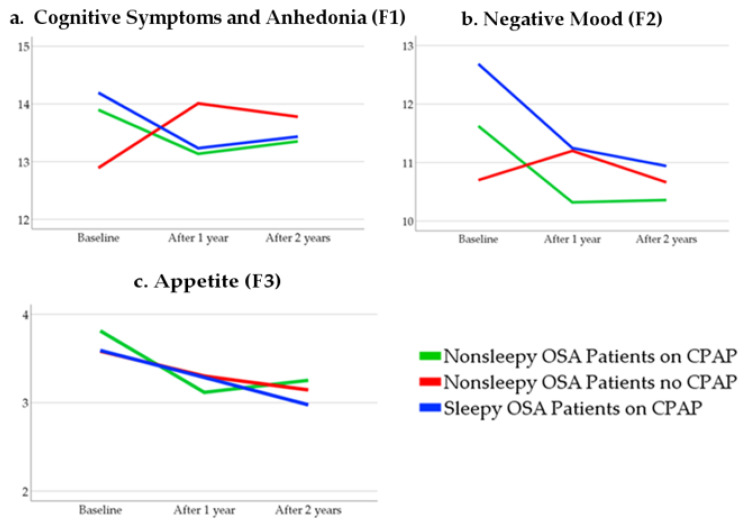
Linear trends (estimated mean values) of the Zung Self-rated Depression subscales over time in the study groups. (**a**). Cognitive Symptoms and Anhedonia (F1); (**b**). Negative Mood (F2); (**c**). Appetite (F3). CPAP, continuous positive airway pressure; OSA, obstructive sleep apnea.

**Table 1 diagnostics-11-01176-t001:** Three-factor structure of the Zung Self-Rating Depression Scale in the study population.

Symptoms	F1	F2	F3
1. I feel down-hearted and blue		0.75	
2. Morning is when I feel the best	0.31		
3. I have crying spells or feel like it		0.53	
4. I have trouble sleeping at night			
5. I eat as much as I used to			0.66
6. I still enjoy sex	0.48		
7. I notice that I am losing weight			0.76
8. I have trouble with constipation		0.31	
9. My heart beats faster than usual		0.49	
10. I get tired for no reason		0.55	
11. My mind is as clear as it used to be	0.70		
12. I find it easy to do the things I used to	0.70		
13. I am restless and can’t keep still		0.60	
14. I feel hopeful about the future	0.61		
15. I am more irritable than usual		0.63	
16. I find it easy to make decisions	0.66		
17. I feel that I am useful and needed	0.64		
18. My life is pretty full		0.59	
19. I feel that others would be better off if I were dead		0.53	
20. I still enjoy the things I used to do	0.60		

F1, Cognitive Symptoms and Anhedonia; F2, Negative Mood; F3, Appetite.

**Table 2 diagnostics-11-01176-t002:** Baseline characteristics of the study population.

	Sleepy OSA on CPAP *n* = 155	Nonsleepy OSA on CPAP *n* = 122	Nonsleepy OSA no CPAP *n* = 122
Age, yrs	61.9 (46–78)	66.1 (40–84)	67.3 (43–81)
Male sex, %	89.0	82.0	86.1
BMI, kg/m^2^	29.0 (22–45)	27.9 (20–42)	28.6(20–39)
Obesity *, %	41.3	27.9	27.9
PCI, %	74.2	73.0	73.0
Current smoking, %	17.4	18.0	13.9
Diabetes, %	25.2	27.9	20.5
Stroke, %	3.9	9.0	10.7
Lung Disease, %	9.7	3.3	9.8
ESS score *	11 (10–23)	6 (1–9)	5 (2–9)
AHI, events/h *	27.7 (12–96)	23.7 (15–69)	24.8 (15–87)
ODI, events/h *	14.7 (0–89)	13.4 (0–64)	11.7 (0–55)
Zung SDS score	43.8 (26–75)	41.3 (25–89)	40.0 (25–66)

AHI, apnea–hypopnea index; BMI, body mass index; ESS, Epworth Sleepiness Scale; ODI, oxygen desaturation index; OSA, obstructive sleep apnea; PCI, percutaneous coronary intervention; SDS, Self-rating Depression Scale. * *p* < 0.05.

**Table 3 diagnostics-11-01176-t003:** Parameters of the Null Model.

Estimates of Fixed Effects					
Parameter	Estimate	Std. Error	95% Confidence Interval		*p*
			Lower Bound	Upper Bound	
**Intercept**					
Sleepy OSA on CPAP	10.20	0.19	9.82	10.59	<0.001
Nonsleepy OSA on CPAP	5.21	0.20	4.79	5.62	<0.001
Nonsleepy OSA no CPAP	5.45	0.21	5.04	5.87	<0.001
**Estimates of Covariance Parameters**					
**Repeated Measures**					
Sleepy OSA on CPAP	9.91	0.88	8.32	11.81	<0.001
Nonsleepy OSA on CPAP	3.34	0.33	2.75	4.06	<0.001
Nonsleepy OSA no CPAP	3.25	0.32	2.68	3.93	<0.001
**Intercept (Subject = ID)**					
Sleepy OSA on CPAP	2.06	0.76	0.99	4.26	0.007
Nonsleepy OSA on CPAP	3.88	0.68	2.75	5.49	<0.001
Nonsleepy OSA no CPAP	4.13	0.69	2.97	5.75	<0.001

CPAP, continuous positive airway pressure; OSA, obstructive sleep apnea.

**Table 4 diagnostics-11-01176-t004:** The fixed effect of the “time” as linear and quadratic trends in the model.

Estimates of Fixed Effects					
Parameter	Estimate	Std. Error	95% Confidence Interval		*p*
			Lower Bound	Upper Bound	
Sleepy OSA on CPAP					
Intercept	12.22	0.244	11.74	12.70	<0.001
Linear Trend	−3.96	0.53	−5.01	−2.9	<0.001
Quadratic Trend	1.05	0.26	0.53	1.56	<0.001
Nonsleepy OSA on CPAP					
Intercept	5.46	0.24	4.98	5.94	<0.001
Linear Trend	−0.65	0.45	−1.55	0.23	0.15
Quadratic Trend	0.21	0.22	−0.21	0.65	0.32
Nonsleepy OSA no CPAP					
Intercept	5.45	0.24	4.97	5.94	<0.001
Linear Trend	0.05	0.43	−0.81	0.91	0.91
Quadratic Trend	−0.03	0.21	−0.45	0.39	0.88

CPAP, continuous positive airway pressure; OSA, obstructive sleep apnea.

**Table 5 diagnostics-11-01176-t005:** The random effect of the “time” as a linear trend in the model.

Estimates of Covariance Parameters					
Parameter	Estimate	Std. Error	95% Confidence Interval		*p*
			Lower Bound	Upper Bound	
**Repeated Measures (Variance)**	4.90	0.64	3.76	6.28	<0.001
**Intercept + Linear Trend**					
UN (1,1)	2.06	0.91	0.86	4.92	0.024
UN (2,1)	0.36	0.56	−0.74	1.48	0.516
UN (2,2)	1.16	0.58	0.43	3.10	0.046

UN, unstructured covariance matrix.

**Table 6 diagnostics-11-01176-t006:** The fixed effect of between-subjects predictors in the model.

Estimates of Covariance Parameters					
Parameter	Estimate	Std. Error	95% Confidence Interval		*p*
			Lower Bound	Upper Bound	
Intercept	6.21	2.85	0.58	11.84	0.03
F1	−0.06	0.05	−0.15	0.03	0.19
F2	0.27	0.07	0.15	0.41	<0.001
F3	0.19	0.11	−0.01	0.41	0.06
CPAP adj/hrs	−0.12	0.10	−0.29	0.06	0.19
Male	0.73	0.64	−0.53	2.00	0.25
BMI	−0.06	0.05	−0.16	0.03	0.17
Age	0.10	0.03	0.00	0.12	0.04
AHI	−0.01	0.02	−0.05	0.04	0.81
ODI	−0.01	0.02	−0.05	0.03	0.66
Smoking	0.05	0.51	−0.96	1.06	0.92
Hypertension	−0.03	0.40	−0.82	0.75	0.93
History of AF	0.48	0.55	−0.62	1.58	0.39
Diabetes	0.72	0.44	−0.17	1.60	0.11
Stroke	−0.11	0.93	−1.95	1.73	0.90
Lung disease	−0.20	0.63	−1.46	1.10	0.74
PCI	0.30	0.47	−0.63	1.22	0.52
Linear	−2.68	0.70	−4.07	−1.29	<0.001
Quadratic	0.59	0.29	0.00	1.17	0.05

AF, atrial fibrillation; AHI, apnea-hypopnea-index; BMI, body-mess-index; CPAP, continuous positive airway pressure; ODI, oxygen desaturation index; OSA, obstructive sleep apnea; PCI, percutaneous coronary intervention.

**Table 7 diagnostics-11-01176-t007:** The final mix model including the significant predictors.

Estimates of Fixed Effects					
Parameter	Estimate	Std. Error	95% Confidence Interval		*p*
			Lower Bound	Upper Bound	
Intercept	8.98	0.71	7.58	10.38	<0.001
F2	0.26	0.05	0.15	0.36	0.001
Linear	−3.41	0.49	−4.40	−2.41	<0.001
Quadratic	0.87	0.24	0.40	1.36	<0.001
**Estimates of Covariance Parameters**					
Repeated Measures	4.67	0.62	3.60	6.10	<0.001
Intercept (Subject = ID) UN (1,1)	2.16	0.91	0.95	4.94	0.01
UN (2,1)	0.47	0.55	−0.60	1.55	0.38
UN (2,2)	0.87	0.55	0.26	2.97	0.10

UN, unstructured covariance matrix.

## Data Availability

Individual participant data that underlie the results reported in this article can be obtained by contacting the principal investigator of the RICCADSA trial; yuksel.peker@lungall.gu.se.

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
