# Peer review of "Association of Excessive Daytime Sleepiness with the Zung Self-Rated Depression Subscales in Adults with Coronary Artery Disease and Obstructive Sleep Apnea"

_diagnostics, 2021, doi:10.3390/diagnostics11071176_

Round 1

Reviewer 1 Report

The study by Celik et al. is a very well-designed study indicating that CPAP treatment can cause a decline in the ESS to score over two years in both sleepy and non-sleepy OSA patients with CAD and improve a negative mood. This study further supports the importance of CPAP treatment in OSA and its benefits. Overall, the manuscript is well-written and coherent.

Please, see my comments.

  • Modify the abstract by adding 2 sentences about the project before the aims.
  • Modify and edit the introduction by adding at least 2 paragraphs related to OSA, CVD, depression, CPAP. It is worthy to mention that OSA can also induce CAD

Methods

P3 L93: Why use Zung self-rated depression scale and not Depression Anxiety Stress Scale (DASS)? Please explain in the manuscript

Results

P6 180-185: the description does not match Figure 4 line colors. Please revise figure

Discussion

Is too short to discuss in more detail for the reader.

It is unfortunate that the authors did not perform sub-analyses within CPAP adherence or usage. Although it is not one of the main aims of this study, it would have added great value to the area.

Reviewer 2 Report

I Notice that the sleepy patients are more obese. Could this be a factor? They also have more hypoxemia compared to the other groups. 

I do not see compliance data. That also might influence sleepiness and overall improvement. Although the outcome was not focused on compliance, it makes a huge impact on outcomes and therefore warrant analysis and comment on depression scores.

The limitations are adequately summarized.  Since the analyses are mostly related to self reported questionnaires, there is inherent bias in reporting which is the patients perspective. It is imperative that these matters be discussed.

Reviewer 3 Report

Manuscript entitled „ Association of Excessive Daytime Sleepiness with the Zung Self-rated Depression Subscales in Adults with Coronary Artery Disease and Obstructive Sleep Apnea” reports on examines the relationship between subscales of the Zung Self-rated Depression Scale and Epworth Sleepiness Scale in a 2-year follow-up of CPAP treatment in OSA patients.

Cut-off for no OSA at 15 is a little misleading as AHI 5-15 is considered a mild presentation.

The long-term follow-up is a great strength of the study.

The statistical analysis is in-depth. Study offers interesting results.

It would be interesting to further investigate the association of ZSDS and insomnia symptoms in similar group.

Round 2

Reviewer 1 Report

I have no further comments. 

Author Response

Thank you!

Reviewer 2 Report

Thank you for addressing my concerns

Author Response

Thanks a lot!